# Use of Intravenous Pulse Steroids to Treat Allergic Bronchopulmonary Aspergillosis in a Non-Compliant Asthmatic Adolescent

**DOI:** 10.3390/children9020252

**Published:** 2022-02-14

**Authors:** Sara G. Hamad, Mutasim Abu-Hasan, Atqah AbdulWahab

**Affiliations:** 1Department of Pediatric Pulmonology, Hamad Medical Corporation, Doha P.O. Box 3050, Qatar; shamad@sidra.org; 2Department of Pediatric Pulmonology, Sidra Medicine, Doha P.O. Box 26999, Qatar; mabuhasan@sidra.org; 3Department of General Pediatric, Weill Cornell Medicine, Doha P.O. Box 24144, Qatar

**Keywords:** allergic bronchopulmonary aspergillosis (ABPA), asthma, pulse steroids

## Abstract

Allergic bronchopulmonary aspergillosis (ABPA) is an immune-mediated inflammatory airway disease that predominantly affects patients with cystic fibrosis (CF) and, less commonly, patients with asthma. ABPA can lead to irreversible lung injury and bronchiectasis if not treated early and aggressively. Long-term oral steroids are the standard therapy of ABPA. However, it is associated with an increased risk of steroids side effects and possible medication noncompliance. Monthly intravenous pulse methylprednisolone (IV-PS) has been used as an alternative to oral steroids to treat CF-related ABPA with a reportedly similar clinical response and less steroid-related side effects. To our knowledge, the use of IV-PS in asthma-related ABPA has not been previously reported. We report the successful management of asthma-related ABPA in an adolescent using intravenous pulse methylprednisolone in addition to oral itraconazole with no significant steroid-related side effects.

## 1. Introduction

Allergic bronchopulmonary aspergillosis (ABPA) is an immune-mediated inflammatory airway disease that predominantly affects patients with cystic fibrosis (CF) and, less commonly, patients with asthma [1,2]. Long-term oral steroids are the standard therapy of ABPA. However, it is associated with an increased risk of steroid side effects and possible medication noncompliance. Monthly intravenous pulse methylprednisolone (IV-PS) has been used as an alternative to oral steroids to treat CF-related ABPA with a reportedly similar clinical response and less steroid-related side effects. To our knowledge, the use of IV-PS in asthma-related ABPA has not been previously reported.

We report the successful management of asthma-related ABPA in an adolescent using intravenous pulse methylprednisolone in addition to oral itraconazole with no significant steroids related side effects.

## 2. Case Presentation

The patient is a 14-year-old boy with asthma who was admitted with mild right pleuritic chest pain and an increased productive cough of a one-month duration. The cough became productive of yellow-greenish sputum and did not respond to an inhaled bronchodilator. The patient reported no wheezing, shortness of breath, hemoptysis, weight loss, nor fever of chills.

The past medical history was significant for mild persistent asthma since the age of 3 years which required multiple emergency department visits, but no hospital or PICU admissions. He has poor adherence to controller inhaled steroid treatment, especially over the past 3 years. On physical examination, the patient had normal vital signs and normal oxygen saturation (97%) of room air. Chest examination revealed bilateral crackles and bronchial breath sounds on the right side.

A complete blood count (CBC) showed elevated white blood cells (WBC) of (17.4 × 10^9^/L) with high eosinophils (4.8 × 10^9^/L). A C-reactive protein and erythrocyte sedimentation rate (ESR) were slightly elevated at 30 mg/L and 34 mm/h, respectively.

A chest X-ray (CXR) showed right middle lobe consolidation (RML) with bilateral scattered nodular opacities (Figure 1(1-A)). The patient was started on intravenous antibiotics with presumptive diagnosis of bacterial pneumonia. Chest computed tomography (CT) was obtained which showed bilateral central bronchiectasis, mucus impaction and areas of segmental, and sub-segmental atelectasis (Figure 1(1-B,1-C)).

Pulmonary function testing on admission revealed normal expiratory flows and volumes with no significant change after bronchodilator but it was noticed to have a decline of his baseline FEV1% predicted in the previous 6 months (from 86% to 73% (a 13% decline) of around 400 mL prior to current presentation). He had increased airway resistance with normal static lung volumes on a body plethysmography. Exhaled nitric oxide was significantly elevated and measured at 138.7 ppb. A lung clearance index of 2.5% was elevated at 145% and predicted indicating ventilation heterogeneity.

The skin prick test was strongly positive for Aspergillus fumigatus (AF) (Figure 2). Total IgE was high (>5000 kU/L) with a high specific anti-AF IgE (32.70 kUA/L) and high anti-AF IgG antibodies (183 mg/L). The immunological work-up was otherwise negative. The sweat chloride test and the QuantiFERON test were normal.

Flexible bronchoscopy was performed during admission and showed diffuse yellow and thick secretions (Figure 3). Bronchoalveolar lavage (BAL) cytology showed macrophage predominance without eosinophils. BAL was positive for galactomannan with positive culture for Aspergillus fumigatus.

Based on the patient’s clinical, radiological and serologic results, ABPA diagnosis was made as the patient was fitting the diagnostic criteria of ABPA (Table 1) as per modified ISHAM criteria. The patient was treated with intravenous pulse steroids instead of oral steroids due to the history of poor adherence to asthma treatment. The patient received a total of four courses of monthly intravenous methylprednisolone (10 mg/kg/day for 3 days). He was also treated with oral itraconazole for 16 weeks and was kept on his previously prescribed inhaled controller medications with an emphasis on parental supervision.

At the end of treatment, symptoms were resolved completely with no reported steroids-related side effects with over almost a year of follow-up. Spirometry showed significant improvement in an FEV1% predicted of 14% (420 mL). Eosinophil count and total IgE reached a nadir of 0 and 1737 kU/L, respectively.

Repeated chest X-rays (Figure 1(4-A) and chest CT showed improved nodular opacities and a complete clearance of the RML consolidation. However, bilateral bronchiectasis was still present but with no mucus impaction (Figure 1(4-B,4-C).

## 3. Discussion

Aspergillus species are a spore-forming fungus that are found ubiquitously in nature. The inhalation of formed spores, especially Aspergillus fumigatus species, may result in a spectrum of Aspergillus-associated respiratory disorders [1].

These disorders affect susceptible hosts and include three main categories: (1) sensitization/allergic manifestations, (2) saprophytic colonization of the respiratory tract, and (3) invasive disseminated disease [1].

The allergic aspergillosis disorders include Aspergillus-induced asthma (AIA), severe asthma with fungal sensitization (SAFS), and allergic bronchopulmonary aspergillosis (ABPA), which is considered the most described form in this category. Some authors would consider these entities as a continuum with AIA at one end of the spectrum and ABPA at the other [2].

Aspergillus-induced asthma (AIA) is a clinical presentation of asthma caused by an IgE-mediated hypersensitivity reaction to Aspergillus antigens. Previous studies showed significant correlation between skin-test positivity and severity of airflow obstruction, as well as an increased incidence of bronchiectasis in patients with AIA [3].

Recently, severe asthma with fungal sensitization (SAFS) has been described in patients with severe asthma. It is considered a separate entity and diagnosed after the exclusion of ABPA due to clinical and therapeutic implications. The diagnostic criteria for SAFS include severe or poorly controlled asthma; a positive skin-prick test result for fungi (Aspergillus fumigatus, Alternaria alternata, or Cladosporium herbarum) or antifungal specific-IgE of >0.4 kU/L; and a total serum IgE level <1000 kU/L. The distinction between SAFS and ABPA is established by lower total IgE levels in SAFS, absence of bronchiectasis and mucoid impaction; and negative serum precipitins to Aspergillus [4,5].

Allergic bronchopulmonary aspergillosis (ABPA) is an immune-mediated lung disease [6] that predominantly affects patients with cystic fibrosis (CF) and asthma [7,8].

Despite being rare in children, ABPA is more prevalent in children with cystic fibrosis with an estimation of 1–15% and increases with age [9,10]. However, few articles have reported ABPA in asthma [11,12,13,14,15,16]. The largest series of ABPA in children with asthma was reported by Shah et al. who described 42 cases of asthma-related ABPA [17].

Allergic bronchopulmonary aspergillosis is related to a hypersensitivity reaction to inhaled Aspergillus spores, and mainly related to Aspergillus fumigatus (AF). The repeated inhalation of the spores may lead to airway colonization in susceptible patients which can result in an allergic response. The common hypersensitivity reaction is mainly type I (IgE-mediated); however, type III (IgG-mediated immune complex) and type IV (cell-mediated) hypersensitivity reactions have also been reported. Nevertheless, tissue invasion has not been documented to occur [18].

The diagnosis of ABPA remains challenging as consensus on the diagnostic criteria and management protocols are still evolving. In 2013, the International Society for Human and Animal Mycology (ISHAM) has proposed criteria for the diagnosis of ABPA [19]. Recently, the modified ISHAM criteria was proposed and analyzed by Saxena et al. [7]. The modified ISHAM criteria mandated the presence of serum total IgE of more than 500 IU/mL and IgE-specific for Aspergillus fumigatus of more than 0.35 KU/L in the context of asthma diagnosis. Additionally, two of the following must be met: (1) Aspergillus fumigatus-specific IgG more than 27 mg/L, (2) Bronchiectasis on chest CT, and (3) Eosinophil count >500 cells/mL.

Early treatment of ABPA may prevent or slow the progression of lung disease and further deterioration to a more severe permanent lung injury [20,21,22].

The mainstay of the standard treatment of ABPA includes systemic corticosteroids. Oral prednisone [23] are often used with an initial dose of 0.5–2 mg/kg/day for 2–4 weeks followed by a slow reduction over a period of 1–3 months. However, prolonged use of oral corticosteroids may result in several side effects in multiple organs, including glucose intolerance, obesity, growth suppression, and osteoporosis [24,25,26,27]. Prolonged oral steroid therapy is especially challenging for patients with CF due to predisposition to diabetes, osteopenia, and growth retardation.

The adjuvant therapy with itraconazole (200–400 mg/day) serves as antigen suppressant and steroid-sparing agent [23].

The administration of monthly pulses of high dose IV methylprednisolone (IV-PS) is frequently used as an effective alternative therapy in diseases that require prolonged oral steroids such as autoimmune diseases [28,29,30,31] and CF-related ABPA [32,33] in both children and adults with less severe adverse effects compared with prolonged oral steroids. Moreover, pulse steroids were used in the management of asthma-related ABPA in adults [34].

Thomson et al. reported successful outcomes with monthly pulses of high-dose IV methylprednisolone in four children with CF and severe ABPA who relapsed after reduction in the dose of oral steroids and who suffered from serious side effects after prolonged therapy [33]. However, to our knowledge, the use of IV-PS in asthma-related ABPA has not been previously described in children.

Other side effects of high dose pulse steroid therapy were uncommonly encountered and reported in several case reports such as hypertension, infections [35], arrhythmias [35,36], altered sensorium [35,37], and hyperglycemia [37]. Fortunately, none of which were observed in our patient.

Omalizumab, a monoclonal antibody against IgE, was described as an alternative yet expensive therapeutic option for the management of patients with ABPA with relapse or with steroid side effects [38,39,40].

## 4. Conclusions

Monthly pulse steroids may be considered as an alternative therapy in children and adolescents with asthma-related ABPA, especially when compliance is a concern. However, further controlled studies are required to establish the efficacy and safety of pulse steroids as the standard of care protocols.

## Figures and Tables

**Figure 1 children-09-00252-f001:**
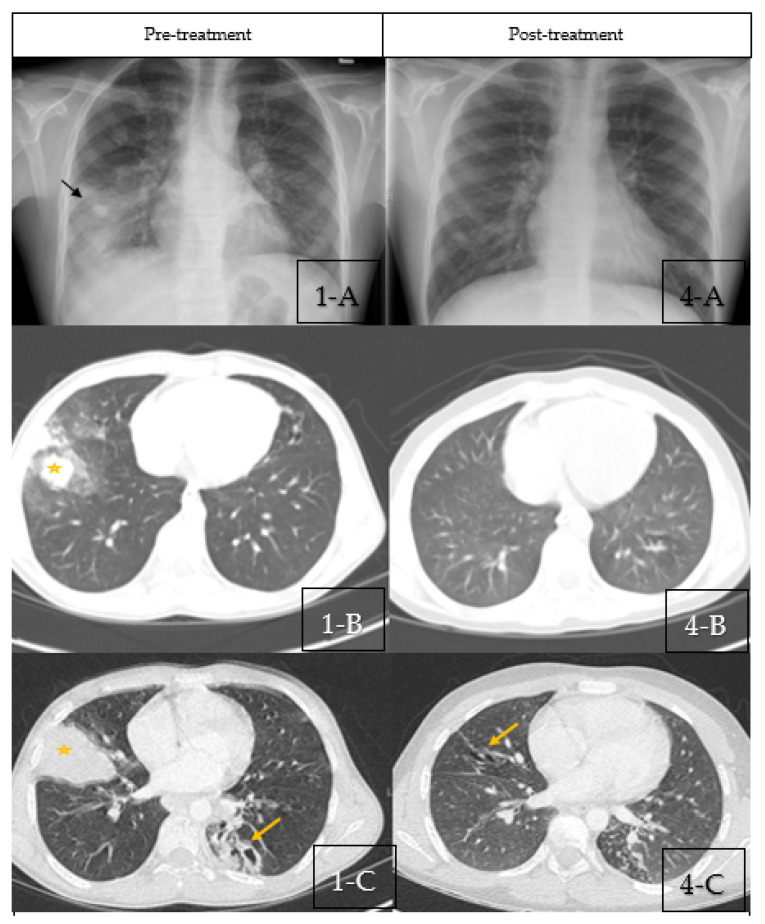
(**1-A**–**1-C**) Pre-treatment Imaging. (**1-A**): Chest X-ray pre-treatment with pulse steroids showing right middle lobe con-solidation (Black Arrow) with bilateral scattered nodular opacities. (**1-B**,**1-C**): Coronal sections of chest CT scan at diagnosis showing bilateral central bronchiectasis, mucus impaction (Yellow Arrow) and areas of segmental, and subsegmental atelectasis (Star). (**4-A**–**4-C**) Post-treatment Imaging: (**4-A**): Chest X-ray post-treatment with pulse steroids showing disappearance of right middle lobe consolidation. (**4-B**,**4-C**): Coronal sections of chest CT scan post-treatment with pulse steroids showing resolution of the previous findings with residual bronchiectasis (Yellow arrows).

**Figure 2 children-09-00252-f002:**
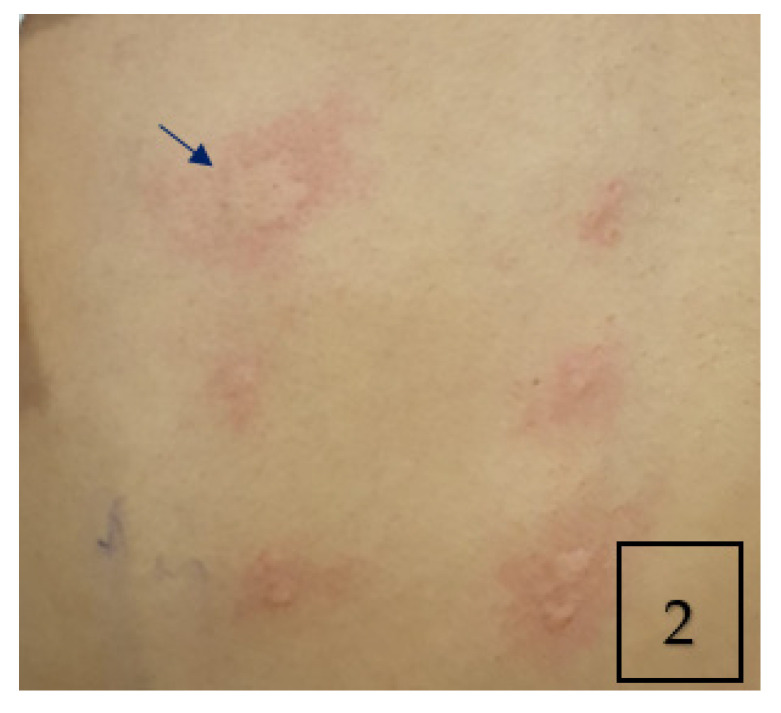
Skin prick allergy test showing positive wheal and flare of 4 × 9 mm in response to aspergillus fumigatus antigen (Arrow).

**Figure 3 children-09-00252-f003:**
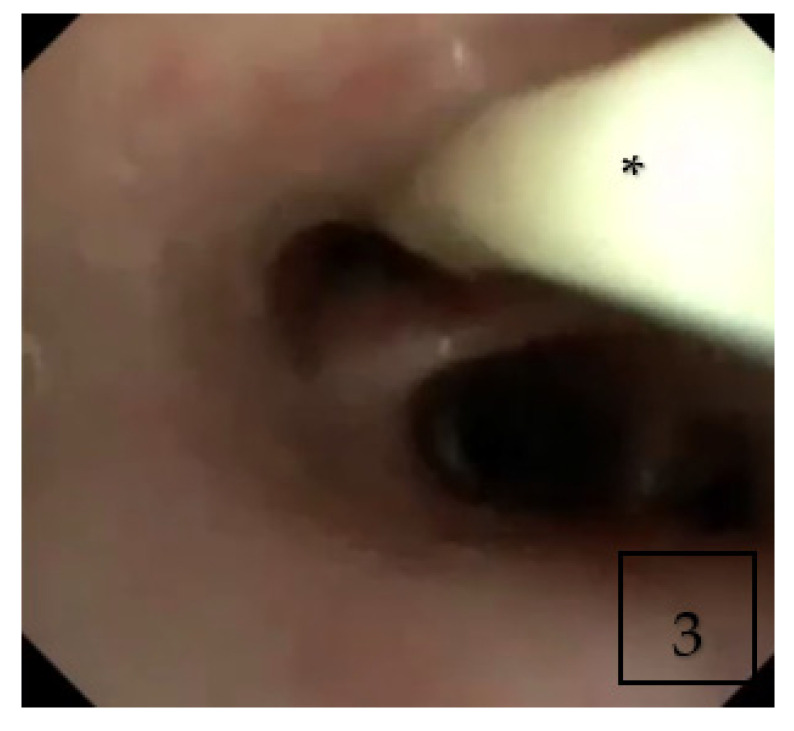
Flexible bronchoscopy showing thick purulent secretions (Asterisk *) in right middle lobe.

**Table 1 children-09-00252-t001:** The patient’s Clinical Symptoms, Pulmonary Function Tests, and Serological Parameters.

	Time of Assessment	Initial Presentation	after 1st Dose of IV-PS	after 2nd Dose of IV-PS	after 3rd Dose of IV-PS	After 4th Dose of IV-PS	at Follow-Up (1–3 Months)	at 1-Year Follow-Up
Clinical symptoms	Cough	Present	Improving	Improving	Resolved	Resolved	Resolved	Resolved
	Chest pain	Present	Improving	Resolved	Resolved	Resolved	Resolved	Resolved
	Appetite	Poor	Poor	Better	Better	Better	Excellent	Excellent
	BMI (kg/m^2^)	16.50	16.50	18.06	19.60	19.10	21.10	18.1
	Weight (Z-score)	−1.7	−1.71	−1.09	−0.76	−0.63	−0.36	−1.25
	Height (Z-score)	−1.39	−1.39	−1.51	−1.60	−1.32	−1.47	−1.51
Pulmonary function tests	FEV1%Pred	82%	92%	97%		97%		94%
	FVC%Pred	84%	97%	96%		98%		95%
	FEV1/FVC	86	86	89		86		87
	FEF25-75%Pred	81%	91%	94%		95%		94%
	Raw%Pred	202%	166%	149%		192%		149%
	RV%Pred	113%	95%	86%		96%		86%
	TLC%Pred	91%	92%	92%		92%		92%
	RV/TLC%Pred	116%	97%	88%		98%		88%
	LCI 2.5%Pred	145%						
	FeNO (ppb)	138.7						28.5
Serology	WBC (×10^9^/L)	17.4	18.1	11.6	23.6	10.9	10.5	11.6
	Eosinophil (×10^9^/L)	4.8	0.9	1.7	0	1.7	2.3	1.7
	Total IgE (kU/L)	>5000	>5000	>5000	3816	1733	1737	1900

BMI: Body mass index; FEV1%pred: Forced expiratory volume in one second %predicted; FVC%Pred: Forced vital capacity %predicted; FEF25-75%Pred: Forced mid-expiratory flow %predicted; Raw%Pred: Airway resistance %predicted; RV%Pred: Residual Volume %Predicted; TLC%Pred: Total Lung capacity %Predicted; LCI 2.5%Pred: Lung clearance index at 2.5%Predicted; FeNO: Fraction of Exhaled nitric oxide; WBC: White blood cells.

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
