# Peer review of "Use of Intravenous Pulse Steroids to Treat Allergic Bronchopulmonary Aspergillosis in a Non-Compliant Asthmatic Adolescent"

_children, 2022, doi:10.3390/children9020252_

Round 1
Reviewer 1 Report
Summary:
This is case study of a single adolescent patient diagnosed with ABPA and treated using a regimen of monthly methylprednisolone given over 3 days. The novelty of this case is that the steroids regimen described has not been described in the paediatric population. There are case reports of its successful application in adults with ABPA. Although ABPA is relatively rare in the paediatric population, It is relevant to the field to develop a treatment regimen that limits the steroid dose and potential side effects associated with a prolonged steroid course. Given the frequent relapse rate seen with this disease, a strength of the case report is that the patient was followed for 1 year without relapse.
General comments:
The patient was worked up appropriately for ABPA and though it can be a difficult disease to confirm (hence the number of different diagnostic guidelines), the diagnosis was adequately confirmed. The treatment regimen was briefly but adequately described and the choice of outcomes measures of treatment success shown graphically by CT and lung function. The discussion section was comprehensive but perhaps more data regarding the successful use of monthly IV methylprednisolone in adults would add to the biologic plausibility of using it in paediatrics populations.
Specific comments:
- there was a formatting problem with all of the figures. The figure number/letter was no longer on the correct figure.
- The figures were presented out of order with the text. They should start at figure 1 and ascent accordingly.
- On Line 37 a value of 73% is given for FEV1 but this does not match the data presented in table 1
- Figure 1 – please indicate on the images and in the figure legend specifically what you want the reader to see and label the different findings with an arrows or asterisks. Consider making the figure a 3x2 panel figure with 1 A-C (pre-treatment) on the top row and 2A-C (post-treatment) on the bottom row
- Figure 4 does not demonstrate the decline in FEV1 which is referred to in the text. It may be better to remove this figure and just have the data presented in the table since other lung function data is also presented. Consider adding a column to the table 1 for 6 months prior to presentation to show that the patient has declined from baseline at diagnosis.
- Figure 5 and 6. This should be 1 figure with 2 panels if it is to share a single figure legend. In the figure legend of figure 5, please explicitly indicate that the weal is in response to aspergillus fumigatus antigen. In Figure 6. Though it may seem obvious, please put a star or an arrow on the mucus.
- On line 80, please indicate/reference which guidelines you used to make the diagnosis of ABPA
- Line 82 – typo – capitalise “the”
- Line 83. Typo - sentence missing the definite article before patient
- Line 85 – refers to 1 year follow up. Can this data be put in Table 1? The table only presents data up to 1-3 months.
Author Response
Dear reviewer,
I want to thank you for the valuable feedback.
The specific comments were all taken into consideration and modified accordingly.
Just to clarify some points.
The patient had normal expiratory flows/volumes upon presentation with an FEV1 of 82% predicted. However, when we reviewed the previous 6-month FEV1-trend, it went down to 73% (This data may be irrelevant to the discussion) but it would help showing the deterioration in PFT prior to presentation.
General comments:
The patient was worked up appropriately for ABPA and though it can be a difficult disease to confirm (hence the number of different diagnostic guidelines), the diagnosis was adequately confirmed. The treatment regimen was briefly but adequately described and the choice of outcomes measures of treatment success shown graphically by CT and lung function. The discussion section was comprehensive but perhaps more data regarding the successful use of monthly IV methylprednisolone in adults would add to the biologic plausibility of using it in paediatrics populations.
Thank you for your valuable comments. The data of using monthly IV steroids is scarce even in adults. The use in CF patients as highlighted in the discussion in adults and pediatric patients aided in the decision making to use IV pulse steroids in our patient.
Specific comments:
- there was a formatting problem with all of the figures. The figure number/letter was no longer on the correct figure.
Figures were captured with legends and numbers so that would no longer be a problem.
- The figures were presented out of order with the text. They should start at figure 1 and ascent accordingly.
I kept the ascent flow through the case description. However, figure 4 to show pre and post treatment with pulse steroids would be out of order, but would work visually better to show the pre and post changes
- On Line 37 a value of 73% is given for FEV1 but this does not match the data presented in table 1
The patient had normal expiratory flows/volumes upon presentation with an FEV1 of 82% predicted. However, when we reviewed the previous 6-month FEV1-trend, it went down to 73% (This data may be irrelevant to the discussion) but it would help showing the deterioration in PFT prior to presentation.
- Figure 1 – please indicate on the images and in the figure legend specifically what you want the reader to see and label the different findings with an arrows or asterisks. Consider making the figure a 3x2 panel figure with 1 A-C (pre-treatment) on the top row and 2A-C (post-treatment) on the bottom row
This is very valid. Modifications were done according to your recommendations.
- Figure 4 does not demonstrate the decline in FEV1 which is referred to in the text. It may be better to remove this figure and just have the data presented in the table since other lung function data is also presented. Consider adding a column to the table 1 for 6 months prior to presentation to show that the patient has declined from baseline at diagnosis.
I removed figure 4. The patient had normal expiratory flows/volumes upon presentation with an FEV1 of 82% predicted. However, when we reviewed the previous 6-month FEV1-trend, it went down to 73% (This data may be irrelevant to the discussion) but it would help showing the deterioration in PFT prior to presentation.
- Figure 5 and 6. This should be 1 figure with 2 panels if it is to share a single figure legend. In the figure legend of figure 5, please explicitly indicate that the weal is in response to aspergillus fumigatus antigen. In Figure 6. Though it may seem obvious, please put a star or an arrow on the mucus.
Done according to your recommendation
- On line 80, please indicate/reference which guidelines you used to make the diagnosis of ABPA
Done according to your recommendation
- Line 82 – typo – capitalise “the”
Done
- Line 83. Typo - sentence missing the definite article before patient
Done
- Line 85 – refers to 1 year follow up. Can this data be put in Table 1? The table only presents data up to 1-3 months
Data was added
Reviewer 2 Report
Interesting case.
I have minor comments.
the authors say that pulses of steroids were preferred because of poor adherence. since also itraconazole for 16 weeks requires good adherence, I wonder if authors checked on that.
asthma treatment with ICS was then back to normal? sputum or cough swab were negative for aspergillus at the end of the treatment?
after ABPA, did he develop severe asthma with fungal sensitisation (SAFS)? the paper might benefit of a small paragraph on the role of aspergillus in asthma (denning dw 2014 clinical and translational allergy).
I would remove Figure 5.
Author Response
Dear Reviewer,
Thank you for your valuable feedback.
The comments were all taken into consideration and were modified in the text. Sputum culture was not repeated due to low yield (BAL is usually the method of acquisition in children, but invasive procedure) and clinical improvement. It is not routinely performed in ABPA.
The addition of SAFS entity enriched the discussion (Our patient fulfilled the criteria of ABPA (SAFS is diagnosed after exclusion of ABPA)).
I wanted to keep figure 5 (Now figure 2) to allow visual presentation of ABPA, it can still be removed if you feel it is not necessary.
I have minor comments.
the authors say that pulses of steroids were preferred because of poor adherence. since also itraconazole for 16 weeks requires good adherence, I wonder if authors checked on that.
Adherence was emphasized and ICS and itraconzole administration was closely supervised by the parents. This was modified in the text.
asthma treatment with ICS was then back to normal? sputum or cough swab were negative for aspergillus at the end of the treatment?
Asthma treatment (ICS) was back to baseline. This was modified in the text. Sputum culture was not repeated due to low yield (BAL is usually the method of acquisition in children, but invasive procedure) and clinical improvement. It is not routinely performed in ABPA.
after ABPA, did he develop severe asthma with fungal sensitisation (SAFS)? the paper might benefit of a small paragraph on the role of aspergillus in asthma (denning dw 2014 clinical and translational allergy).
The addition of SAFS entity enriched the discussion (Our patient fulfilled the criteria of ABPA (SAFS is diagnosed after exclusion of ABPA)).
I would remove Figure 5.
I wanted to keep figure 5 (Now figure 2) to allow visual presentation of ABPA, it can still be removed if you feel it is not necessary.
Thank you for your valuable feedback.